# A Feasibility Study to Increase Chronic Hepatitis C Virus RNA Testing and Linkage to Care among Clients Attending Homeless Services in Amsterdam, The Netherlands

**DOI:** 10.3390/diagnostics11071197

**Published:** 2021-06-30

**Authors:** Ellen Generaal, Hilje Logtenberg van der Grient, Eberhard Schatz, Daniela K. van Santen, Anders Boyd, Sara K. Woods, Bert L. C. Baak, Maria Prins

**Affiliations:** 1Department of Infectious Diseases, Research and Prevention, Public Health Service of Amsterdam, 1018 WT Amsterdam, The Netherlands; dvsanten@ggd.amsterdam.nl (D.K.v.S.); aboyd@ggd.amsterdam.nl (A.B.); mprins@ggd.amsterdam.nl (M.P.); 2De Regenboog Groep, 1013 GE Amsterdam, The Netherlands; logtenberg@lvdg-consultancy.com (H.L.v.d.G.); eschatz@correlation-net.org (E.S.); 3Disease Elimination Programs, Burnet Institute, Melbourne, VIC 3004, Australia; 4Stichting HIV Monitoring, 1105 BD Amsterdam, The Netherlands; 5Mainline, 1052 HN Amsterdam, The Netherlands; s.woods@mainline.nl; 6Department of Gastroenterology and Hepatology, OLVG Hospital, 1091 AC Amsterdam, The Netherlands; L.C.Baak@olvg.nl; 7Department of Infectious Diseases, Amsterdam UMC, Location AMC, Amsterdam Infection and Immunity (AII), University of Amsterdam, 1105 AZ Amsterdam, The Netherlands

**Keywords:** hepatitis C, people who inject drugs, homelessness, test and treat approach, cascade of care

## Abstract

People who inject drugs (PWID) are disproportionately affected by hepatitis C virus (HCV) infections and are frequently homeless. To improve HCV case finding in these individuals, we examined the feasibility of rapid HCV RNA testing in homeless services in Amsterdam. In 2020, we provided a comprehensive service to homeless facilities, which included workshops on HCV for personnel, a “hepatitis ambassador” at each facility, a rapid, onsite HCV RNA fingerstick test service, and assistance with linkage to care. Risk factors for HCV RNA-positive status were examined using Bayesian logistic regression. Of the 152 participants enrolled, 150 (87% men; median age: 47 years) accepted rapid HCV testing. Seven tested HCV RNA positive (4.7%, 95%CrI = 1.31–8.09; 7/150). Of these, five (71%) were linked to care, of whom four (57%, 4/7) initiated treatment and one (14%, 1/7) delayed treatment due to a drug–drug interaction. Of these four people, two completed treatment (50%), of whom one (25%) achieved sustained virologic response after 12 weeks. HCV RNA-positive individuals were more likely to originate from Eastern Europe (posterior-odds ratio (OR) = 3.59 (95% credible interval (CrI) = 1.27–10.04)) and to inject drugs (ever: posterior-OR = 3.89 (95% CrI = 1.37–11.09); recent: posterior-OR = 3.94 (95% CrI = 1.29–11.71)). We identified HCV RNA-positive individuals and linkage to care was relatively high. Screening in homeless services with rapid testing is feasible and could improve HCV case finding for PWID who do not regularly attend primary care or other harm reduction services for people who use drugs.

## 1. Introduction

Hepatitis C virus (HCV) is a major cause of liver cirrhosis and hepatocellular carcinoma [1]. An estimated 50–80% of adults with acute HCV infection develop chronic infection. Of the people chronically infected with HCV, 12–30% develop liver cirrhosis within 20 years if left untreated [2]. As part of the HCV elimination goals established by the World Health Organization (WHO), 90% of all individuals infected with HCV should be diagnosed by 2030 [3]. In 2015, only 20% of HCV-infected people were estimated to be diagnosed worldwide [4]. A study in 2012 from the southern region of the Netherlands indicated that 66% of HCV infections were ‘hidden’ to current screening practices [5]. Nevertheless, coverage of HCV diagnoses may have improved with recent HCV screening programs, such as increased case finding among PWID [6,7].

The estimated overall prevalence of chronic hepatitis C infections in the Netherlands is low (0.16% in 2016) [8]. However, prevalence estimates are much higher in men who have sex with men (4.9% for HIV positive MSM), first generation migrants from HCV endemic countries (0.9%), and current or former people who inject drugs (PWID) (59%) [8]. As such, the Dutch Health Council has recommended case finding for these HCV key populations, rather than for the general population [9].

At the present moment, current and former PWID engaged in harm reduction (HR) programs and MSM living with HIV or on preexposure prophylaxis (PrEP) are regularly screened for HCV infection in the Netherlands. PWID are also given testing opportunities for HCV as part of being included in research and demonstration projects [6,7,10,11]. However, there are some populations that might not receive adequate access to HCV testing. For instance, PWID who are homeless might not frequently access HR reduction program services or primary care, without which opportunities for HCV testing could be missed. People who are homeless in Amsterdam, including migrants, often use temporary housing services (“maatschappelijke opvang”), where HCV testing is not routinely offered. Even if homeless individuals have been tested for HCV in the past and had a positive result, they might not yet be linked to care or even aware of their chronic infection status [7].

An earlier HCV testing study pilot in 2017 among 225 clients of homeless centers in Amsterdam [12] showed high dropout rates between initial HCV rapid antibody test (OraQuick) and confirmatory HCV RNA testing (32%, 7/22) and between positive HCV RNA result and treatment initiation (67%; 4/6 did not start treatment) [12]. To improve the HCV “cascade of care”, recommendations from that study and others [13] were to: (1) educate personnel of homeless services about disease progression, risk factors, and treatment related to HCV, (2) assign one coordinating “hepatitis ambassador” (social care worker) to each center to stimulate participation in HCV screening, (3) conduct onsite rapid HCV RNA testing to prevent dropout, and (4) assign dedicated hospital staff to assist HCV-positive individuals into clinical care [12].

The aim of the current project was to assess the feasibility of onsite rapid HCV RNA testing and linkage to care among clients attending temporary homeless services in Amsterdam, the Netherlands, while implementing earlier recommendations for onsite testing and support of hospital staff to improve testing and treatment uptake.

## 2. Materials and Methods

### 2.1. Design, Homeless Services, and Participants

This cross-sectional study was conducted during January–March 2020 by a De Regenboog Groep foundation in collaboration with the Public Health Service of Amsterdam and the harm reduction foundation, Mainline. The eight homeless services participating were part of the following foundations: De Regenboog Groep (location AMOC), The Salvation Army (Leger des Heils; Middelenbewust, Gastenburgh, and Zuiderburgh locations), HVO Querido (De Vaart and Anton de Komplein locations), PerMens (formerly Volksbond, Jan Rebelstraat location) and the MDHG union for people who use drugs (PWUD.) Some centers included were low-threshold homeless daytime services, available on a drop-in basis (AMOC and MDHG). AMOC provides limited overnight shelter and is mainly visited by people from Eastern Europe. All the other locations provide short-term (up to 3 months) and/or long-term (more than 3 months) overnight shelter for people from various backgrounds with a Dutch residency permit. This study is a continuation of a previously conducted project in 2017, in which barriers to HCV testing and care were addressed in centers providing homeless services. The eight homeless services represented in this study were selected based on expert opinion indicating that many PWID attend these services. These centers either did not participate in or recruited a small number of participants for the previous project [12].

### 2.2. Ethical Procedures

All participants provided written informed consent. The project was conducted according to the ethical guidelines of the 1975 Declaration of Helsinki. The local medical ethics committee of the OLVG hospital in Amsterdam approved the screening and testing project (non-WMO declaration, ACWO 19.104).

### 2.3. Procedures to Increase HCV Testing and Linkage to Care

We implemented the following recommendations from the previous project conducted in 2017 [12]: (1).Educating employees about HCV

Personnel of the eight selected social care centers were invited to participate in a workshop with information related to HCV: modes of transmission, key populations, consequences of HCV infection, prevention, and current treatment options. A total of 81 employees participated in the eight workshops (one workshop per center). 

(2).Instating hepatitis ambassadors at homeless services

We selected one employee per center who was familiar with and trusted by clients to be appointed the “hepatitis ambassador” for the project. The hepatitis ambassador was responsible for adequately communicating study procedures and test results with the clients during the study, and for linking HCV-positive clients to the physician assistant of the hospital. 

(3).Rapid onsite HCV testing

We performed onsite HCV RNA testing using a validated blood fingerstick assay (Xpert HCV Viral Load assay, Cepheid, Sunnyvale, CA, USA) [14]. In a previous study, the sensitivity and specificity of this assay to detect HCV RNA in samples collected by fingerstick were 95.5% (95% CI 84.5–99.4) and 98.1% (95% CI 93.4–99.8), respectively [15]. Test results were offered within 60 min to ensure HCV RNA test results were received and individuals could be linked to care (see data collection). Assays (at €90 per test) and the testing device were provided free-of-charge by the supplier for the duration of this project.

(4).Improving linkage to care for HCV-positive individuals

The project coordinator (HLvdG) supervised the test results. If a client tested HCV positive, the project coordinator contacted a previously appointed physician assistant from the hepatitis department of the OLVG hospital. The physician assistant contacted the hepatitis ambassador to schedule an appointment with the client, which took place at the homeless service. During this consultation, follow-up actions, including a treatment plan, were discussed with the client and the hepatitis ambassador. Given the hard-to-reach population, we decided to prescribe DAA treatment if the patient reported no contra-indications at intake. DAA treatment included an 8-week course of glecaprevir/pibrentasvir. After 6 weeks of treatment, clients underwent liver function testing, an ultrasound, and a transient elastography measurement (TEM; FibroScan^®^, EchoSens, Paris, France) at the hospital. After end of treatment (EOT) and 12 weeks after EOT, HCV RNA was tested at the hospital. A negative HCV RNA at 12 weeks after EOT defined sustained virologic response (SVR12). Treatment was paid through the client’s health insurance or a special arrangement for undocumented migrants (CAK arrangement, GGD GHOR, The Netherlands). Follow-up care for HCV-positive individuals was monitored by the project coordinator.

### 2.4. Recruitment and Data Collection Procedures

Personnel were asked to actively recruit clients of the homeless centers, with the addition of posters and fliers as prestudy announcements. Clients participated voluntarily and received a small gift as an incentive. Participants were asked to fill out informed consent and a questionnaire with questions on sociodemographic characteristics (age, gender, and country of birth), medical history related to HCV and HCV-related risk factors, and sexual behavior. Subsequently, blood samples were drawn with the use of a fingerstick, from which HCV RNA was assessed using the Xpert^®^ HCV VL Fingerstick assay (Cepheid; [15]). Participants received test results within 60 min. 

### 2.5. Sociodemographic Characteristics, HCV-Related Risk Factors, and Sexual Behavior

Sociodemographic characteristics included age, gender, and country of birth. In line with previous research [12,16,17], HCV-related risk factors included injecting drug use (ever or recent; <30 days); ever sharing needles or syringes, straws or crack pipes, or razors or tooth brushes; medical procedures in HCV endemic countries (i.e., blood transfusion, tattooing, piercing, and/or dental procedures); or tattooing/piercing in a nonregistered location in the Netherlands. Sexual behavior included condomless anal sex reported by men and ever sharing sex toys.

### 2.6. Statistical Analyses

Descriptive baseline characteristics were reported as means, median (for age), or percentages. Bayesian logistic regression was used to examine sociodemographic characteristics, HCV-related risk factors, and sexual behavior in relation to HCV-positive status. Since few cases of HCV were expected, we suspected that parameter estimates from conventional regression techniques could be overinflated and uncertain (i.e., sparse data bias) [18]. To minimize this bias, we employed a penalized regression approach in which estimates from the sample were pulled towards more realistic ones assumed from prior knowledge [19]. For each risk factor, we assigned a prior distribution of the odds ratio (OR) based on the strength of association, as anticipated from previous investigation and expert consensus [20]. We assumed a noninformative prior for the intercept (i.e., uniform prior bound from −10,000 to 10,000). Using these priors together with the data, a posterior distribution of ORs was estimated with Markov Chain Monte Carlo methods using the “bayes” prefix commands in STATA. The median of this distribution defined the parameter estimate (i.e., “posterior-OR”) and their 2.5% and 97.5% quantiles defined the 95% credible interval (CrI). Both prior-OR and posterior-OR are provided, as the interpretation of posterior-OR depends on the prior-OR. Statistical analyses were carried out using SPSS (v26.0, IBM Corp, Armonk, NY, USA) and STATA (v15.0, College Station, TX, USA).

### 2.7. Sensitivity Analyses

Given that the posterior-ORs are influenced by the prior distributions, we repeated the analysis using (1) noninformative priors, and (2) a different prior OR distribution with weaker effect sizes for posterior-ORs whose 95% CrI did not include 1.

## 3. Results

### 3.1. Characteristics of the Study Population

A description of the study sample is provided in Table 1. A total of 152 clients from these eight locations participated in the current project. Of the 152 participants included, 150 (87% men; median age: 47 years, interquartile range (IQR) = 38–56) accepted blood draw and subsequent testing. For two of the 152 people who initially agreed to participate in the study, blood draw was unsuccessful due to calluses on their fingers, and they refused further testing (Figure 1). Seven people (of *n* = 32) reported that they had tested positive for HCV in the past (1985–2019), two of whom tested HCV RNA positive in our study.

### 3.2. Characteristics of the HCV RNA-Positive Individuals

Of the 150 people with a successful blood draw, seven tested HCV RNA positive (4.7% from total of 150 people, 95% CrI = 1.31–8.09) and were found in two test locations in Amsterdam, a long-term shelter and a drop-in center. HCV-positive individuals were born in Eastern Europe (Former Soviet Union, *n* = 5) or The Netherlands (*n* = 2). HCV RNA-positive individuals had a median age of 33 years (IQR = 29–59) and 86% (6/7) were men. The proportion of HCV RNA positivity among PWID in our study was 18%. Of all HCV RNA-positive individuals, 57% (4/7) reported ever injecting drug use, and 43% (3/7) reported recent injecting drug use. These proportions were higher than those of the overall study population (ever: 15%, 22/146; recent: 6.7%, 10/149). Five people were unaware of their HCV status (Table 1) and two people previously tested positive for HCV. One person declared that he had tested HCV positive in the past, but was never treated (for unknown reasons). The other person declared that he was not treated in 2010 because of injecting drug use. The medical specialist of the hospital likely advised against treatment with interferon at that time. Two people unaware of their HCV status reported recent and prior injecting drug use. Two people reported no recent or prior injecting drug use, and one person did not answer this question during the interview.

### 3.3. Linkage to Care

Of all seven HCV RNA-positive individuals, five were followed up in care (71%). Two HCV RNA-positive individuals did not return after the first COVID-19 lockdown (March 2020) and could not be contacted by personnel from the homeless service or the hospital. Of the five people who were followed up, four started DAA treatment (57%; completed, *n* = 2; ongoing, *n* = 1, lost to follow-up, *n* = 1; see Figure 1). For one person, DAA treatment could not be started due to a potential drug–drug interaction with his anti-epileptic treatment (levetiracetam). The participant did not wish to contact his medical specialist to change his treatment regimen. Three people started DAA treatment in September 2020 (7 months after testing), two of whom completed treatment and one was lost to follow-up (he was incarcerated). One person initially refused treatment, but started in February 2021 (12 months after testing). Of the two people who completed treatment, SVR12 was assessed in one individual. For the other, end of treatment (EOT) testing was delayed to 6 weeks after EOT due to limited testing during the coronavirus crisis. We were unable to contact the participant for an appointment 12 weeks after EOT; however, an appointment is scheduled in September 2021 to assess SVR12. For both of these individuals, HCV RNA was negative at their last assessment. During routine testing, this same participant showed suspicion of liver fibrosis based on the ultrasound and TEM; therefore, follow-up testing of liver function was also scheduled in September. The other patient showed no signs of liver damage during routine testing.

### 3.4. Risk Factors for HCV RNA Positivity

Table 2 shows the results from the Bayesian regression analyses examining risk factors for HCV RNA positivity. HCV RNA positivity was associated with coming from an Eastern European country (posterior-OR = 3.59, 95% CrI = 1.27–10.04), ever injecting drug use (posterior-OR = 3.89, 95% CrI = 1.37–11.09), and recent injecting drug use (posterior-OR = 3.94, 95% CrI = 1.29–11.71). Ever sharing needles or syringes was also associated with HCV RNA positivity (posterior-OR = 3.16, 95% CrI = 0.96–9.39), but the 95% CrI crossed 1. The few numbers of infections precluded us from conducting multivariable analyses. The same factors were associated with HCV RNA positivity when using noninformative priors, although with increased uncertainty (i.e., wider CrI) (Appendix A), or when using priors with weaker effects (Appendix A).

## 4. Discussion

This cross-sectional study examined the feasibility of a rapid, onsite HCV RNA screening test and the resulting linkage to care for 150 clients of eight homeless services in Amsterdam, the Netherlands. We found a higher proportion of chronic hepatitis C (HCV RNA positive) infection among participants, at 4.7%, than is expected for the general population in the Netherlands (0.16%; [8]). An earlier pilot study among clients of homeless services in Amsterdam showed a comparable proportion with positive HCV RNA tests of 4.4% (10/225) [12]. Previous estimations of HCV prevalence in homeless PWID across Europe are commonly based on HVC antibodies and range from 35% to 70% [21]. One cohort study conducted in Europe among homeless individuals, prisoners, and PWID showed a HCV RNA prevalence ranging from 9% to 43% [22]. The proportion of HCV RNA positivity among PWID in our study is 18%, which is somewhat lower compared to that of PWID in the European Union region, which ranges between 26% and 90% (Studies 8, 16–19 in: [23]). In the Netherlands, the proportion of PWUD with a chronic HCV infection appears to have decreased over time from 31% in 2005 to 18% in 2015, but remains substantial, as reported by the Amsterdam Cohort Studies [10]. The estimated proportion of ever chronically infected PWID in the Netherlands was estimated at 59% in 2016 [8].

Offering a rapid onsite HCV RNA fingerstick test was successful, as it prevented potential dropout between antibody test and HCV RNA testing. Dropout was a major issue in the earlier pilot study in Amsterdam [12]. Onsite point-of-care HCV RNA testing is also in line with WHO recommendations for improving HCV-related interventions [13] and other studies [24,25,26]. HCV antibody testing is commonly followed up with HCV RNA testing for those who are antibody-positive; the reasoning being that HCV RNA testing (at ±€90 per test) would be more costly if conducted on all individuals rather than on those who first test positive for HCV antibodies (at ±€30 per test). However, point-of-care HCV RNA assays may be worthwhile for this key population to enhance testing uptake and linkage to care.

In the current study, five of the seven HCV RNA-positive people were unaware of their status. Linkage to care was relatively good (71%, 5/7). In an earlier pilot study in homeless centers (67%; [12]) and in studies from the USA, linkage ranged from 66% for inpatient and 58% for outpatient populations to 23% for people visiting the emergency department. This finding suggests that the guidance provided by the hepatitis ambassadors and hospital staff was successful, thereby supporting earlier recommendations [13,27]. A qualitative study among homeless people in the USA has suggested that a designated HCV coordinator, an incentive for clients to continue follow-up appointments, reduced treatment periods (to a maximum 2 months), education of personnel and HCV-positive patients, and peer support for HCV patients are needed to promote linkage to care [27]. Unstable accommodation in particular is a major barrier for HCV treatment uptake [28,29,30], which could be overcome by extending shelter stays for HCV-positive clients [27,31].

Despite the fact that HCV testing concurred with the beginning of the coronavirus crisis (March 2020), treatment was successfully initiated for 57% (4/7) of the HCV RNA-positive individuals, suggesting acceptable levels of referral to care and adherence. These proportions were comparable to the previous pilot study (58%; [12]). Nevertheless, three people had already finished treatment in January 2021 and the first COVID-19 lockdown might have contributed to two other HCV RNA-positive individuals being lost to care in our study. Linkage to care is currently an even greater challenge; for example, due to limited access to hospital care during periods of extensive lockdown measures and psychological barriers to visiting the hospital during the COVID-19 pandemic.

Our study shows that individuals originating from Eastern European countries and who were formerly or currently injecting drugs were at high risk for HCV RNA positivity. This is in line with the high prevalence estimates of chronic HCV infections found in PWID [8] and in former Soviet Union countries (2.7%; [32]), in particular among PWID living in these regions (>50% anti-HCV positive; [33]). Countries in these regions were among the top three with the highest net migration rate (The net migration rate is the difference between the number of mmigrants (people coming into an area) and the number of emigrants (people leaving an area) throughout the year) to the Netherlands in 2019 [34]. There have been several HCV screening studies for former PWID in the Netherlands [6,7,10,11], but HCV case finding could also be offered in the temporary homeless services on a regular basis. Recommendations to adequately reach this group may include developing a standard HCV screening protocol for homeless services and providing training to personnel to increase awareness and normalize testing in these settings.

Our study has some limitations. First, included participants were more likely opiate injectors in socioeconomically difficult situations; therefore, our findings cannot be generalized to other groups of PWID or PWUD. Second, we did not register the number of people who were unwilling to participate and their reasons for nonparticipation. Third, there were few HCV RNA-positive individuals and hence there was likely insufficient power to identify other determinants of HCV RNA positivity. Conversely, ORs from highly common factors found within the low numbers of individuals with an HCV RNA-positive test could have been inflated (Appendix A). We attempted to mitigate this bias by using a Bayesian logistic regression model and performing sensitivity analyses on prior assumptions; however, a falsely positive association cannot be completely ruled out. Furthermore, some questionnaire items, such as anal sex with male partners, were commonly missing, which could have induced a differential reporting bias. Fourth, we cannot rule out social desirability bias from the reporting of our study group. Lastly, we cannot generalize our results to other countries or settings, although we did purposely include participants with various backgrounds and from different types of shelters in Amsterdam. Nonetheless, this study was successful in testing for HCV RNA and linking individuals to care in a population experiencing homelessness with high rates of injecting drug use, which has been traditionally viewed as hard-to-serve in standard harm reduction programs and addiction care.

Our study shows that for HCV case finding, onsite rapid HCV RNA testing and the support of hospital staff may be relevant to more appropriately serve people experiencing homelessness, including current or former PWID. Future testing interventions could focus on people of Eastern European origin. A one-step screening strategy based on HCV RNA detection could engage these groups into care and contribute to HCV elimination goals set by WHO to scale-up testing and treatment for HCV key populations [35].

## Figures and Tables

**Figure 1 diagnostics-11-01197-f001:**
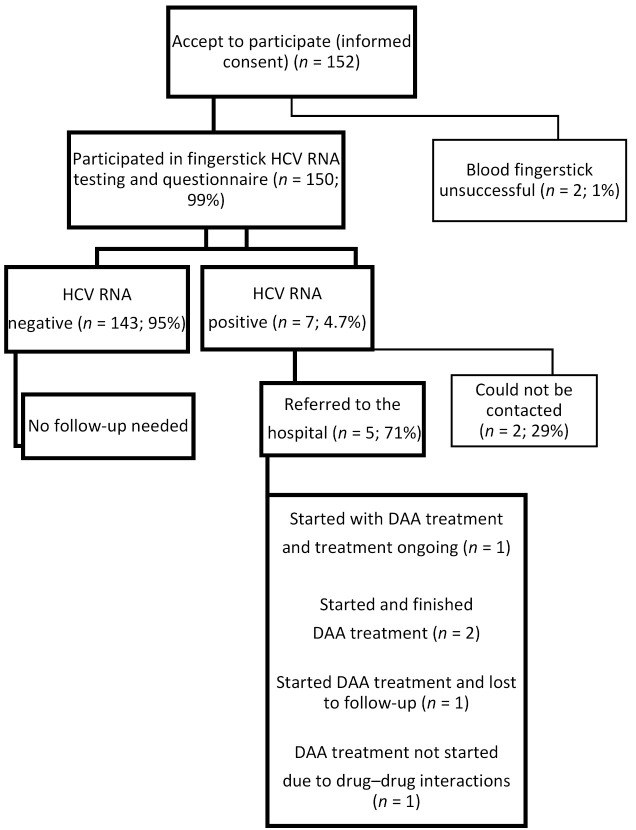
Flowchart for recruitment, study procedures, and linkage to care in eight homelessness services in Amsterdam, the Netherlands, January–March 2020.

**Table 1 diagnostics-11-01197-t001:** Characteristics of the study population of eight homeless services in Amsterdam, the Netherlands, in January–March 2020 (*N* = 152).

	***N***	**Percentage or Median (IQR)**	**Total** ***N ****
Sociodemographic characteristics			
Male sex	132	87%	151
Age in years		47 (38–56)	152
Continent/country of birth			152
Western Europe, The Netherlands	41	27%	
Western/Northern Europe (other)	8	5.3%	
Eastern Europe (Former Soviet Union)	26	17%	
Africa	34	22%	
South America, Suriname	22	15%	
North America, USA	1	0.7%	
Caribbean/Antilles	14	9.2%	
Central Asia (Middle East)	6	3.9%	
	***N***	**Percentage or Median (IQR)**	**Total** ***N ****
Current treatment			
Methadone treatment, yes	25	17%	149
Homeless service/test location in Amsterdam			152
Long-term shelter	68	45%	
Short-term shelter	30	20%	
Mixed center (short- and long-term)	17	11%	
Drop-in center	37	24%	
HCV-related risk factors			
Procedures in HCV endemic country (lifetime)			
Medical procedure	16	11%	146
Blood transfusion	4	2.8%	144
Tattoo or piercing	29	20%	146
Dental procedures	23	16%	144
Procedures in the Netherlands (lifetime)			
Tattoo/piercing in a nonregistered location	26	18%	142
Injecting drugs (ever)	22	15%	146
Injecting drugs (last 30 days)	10	6.7%	149
Ever sharing of:			
needles and/or syringes	7	4.8%	146
straws and/or crack pipe	70	48%	146
razors and/or toothbrushes	35	25%	140
Sexual behavior			
Condomless anal sex by men (ever)	15	13%	120
Ever sharing sex toys	11	8.1%	136
Medical history related to HCV			
Had been previously tested for HCV (1985–2019) **	38	66%	110
Had previously tested positive for HCV	7	22%	32
HCV RNA testing			
Participated in HCV RNA testing			152
Yes	150	99%	
No, fingerstick unsuccessful ***	2	1.3%	
Result HCV RNA test			150
Negative	143	95%	
Positive	7	4.7%	
Unaware of HCV RNA positivity	5	71%	7

* Total *N*-value differs due to missing values on individual measures. ** Medical history was based on self-report, meaning that prior HCV testing could be based on both HCV antibody or RNA testing. *** Due to calluses on fingers. Short-term shelter concerns stays lasting a maximum of 3 months (Gastenburgh, Total *n* ± 70). Long-term shelters concern stays for longer than 3 months (Middelenbewust, Total *n* ± 45; De Vaart, Total *n* ± 36; Zuiderburgh, Total *n* ± 80; Jan Rebelstraat, Total *n* ± 75). Mixed shelters concern both short- and long-term stays (Anton de Komplein, Total *n* ± 55). Drop-in centers are AMOC (Total *n* ± 70) and MDHG (Total *n* ± 50). Abbreviations: HCV, hepatitis C virus; SD, standard deviation; IQR, interquartile range.

**Table 2 diagnostics-11-01197-t002:** Characteristics and risk behaviors according to hepatitis C virus (HCV) RNA status, and prior and posterior estimates of risk factors for HCV RNA positive status.

	HCV RNA Negative (*N* = 143), Median (IQR) or Number/N (%)	HCV RNA Positive (*N* = 7), Median (IQR) or Number/N (%)	Prior OR (95% CrI)	Posterior OR (95% CrI)
Sociodemographic characteristics				
Age (per 10 year increase) ^a^	47 (39–56)	33 (29–59)	0.90 (0.23–3.60)	0.67 (0.40–1.12)
Females (versus males)	18/142 (12.7)	1/7 (14.3)	1.00 (0.25–4.00)	1.02 (0.30–3.24)
Eastern European origin	20/143 (14.0)	5/7 (71.4)	1.50 (0.38–6.00)	3.59 (1.27–10.04)
HCV-related risk factors				
Injecting drug use (ever)	18/138 (13.0)	4/6 (66.7)	2.00 (0.50–8.00)	3.89 (1.37–11.09)
Injecting drug use (last 30 days)	7/141 (5.0)	3/6 (50.0)	2.00 (0.50–8.00)	3.94 (1.29–11.71)
Ever sharing needles or syringes	5/138 (3.6)	2/6 (33.3)	2.00 (0.50–8.00)	3.16 (0.96–9.39)
Ever sharing straws/crack pipe	65/138 (47.1)	4/6 (66.7)	1.50 (0.38–6.00)	1.69 (0.56–4.95)
Ever sharing razors/toothbrush	35/134 (26.1)	1/5 (20.0)	1.50 (0.38–6.00)	1.47 (0.38–3.78)
Procedures ^b^ in HCV endemic country	46/138 (33.3)	1/6 (16.7)	1.50 (0.38–6.00)	1.00 (0.32–2.81)
Tattoo/piercing in nonregistered location ^c^	25/134 (18.7)	1/7 (14.3)	1.50 (0.38–6.00)	1.10 (0.35–3.09)
Sexual behavior				
Condomless anal sex	16/130 (14.0)	0/4 (0)	1.50 (0.38–6.00)	1.05 (0.29–3.47)
Sharing of sex toys	11/114 (8.5)	0/5 (0)	1.50 (0.38–6.00)	1.14 (0.31–3.80)

^a^ Age was centered at 18 years; ^b^ includes any medical procedure, blood transfusion, tattooing, piercing, and/or dental procedures; ^c^ taking place in the Netherlands. CrI, credible interval; HCV, hepatitis C virus; OR, odds ratio; IQR, interquartile range.

## Data Availability

The data presented in this study are available on request from the corresponding author.

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
