# Peer review of "A Feasibility Study to Increase Chronic Hepatitis C Virus RNA Testing and Linkage to Care among Clients Attending Homeless Services in Amsterdam, The Netherlands"

_diagnostics, 2021, doi:10.3390/diagnostics11071197_

Round 1

Reviewer 1 Report

Generaal et al. assessed the feasibility of rapid test to early detect HCV viremia carriers and to increase the proportion of carriers to link to care in homeless services in Netherland. The manuscript was well-written and did provide an efficient model for HCV elimination. Several minor comments are raised for further clarification.

  1. Once clients were identified to have HCV viremia and were linked to care, please state how long were needed in the 4 patients to receive DAAs?
  2. Did all viremic patients receive baseline routine testing (hematological, biochemical etc…) before initiating DAAs? Because some DAAs are contraindicated for patients with decompensated liver diseases or not suitable for active HCCs, pre-treatment full assessments are needed for all treated patients. Can the authors provide the disease status for all the viremic patients and the information about the use of DAAs (which kind of DAAs, treatment duration)?
  3. Why did the patients receive testing after 6 weeks of treatment because it was not usual for this assessment? Please clarify it.
  4. Why did the patients receive HCV RNA testing after 8 weeks of treatment in the Methods? It was not correct to assess SVR during treatment. The testing should be done at 12 weeks off therapy. The illustration in Section 3.3 should also be clarified.

Author Response

Generaal et al. assessed the feasibility of rapid test to early detect HCV viremia carriers and to increase the proportion of carriers to link to care in homeless services in Netherland. The manuscript was well-written and did provide an efficient model for HCV elimination. Several minor comments are raised for further clarification. 

1. Once clients were identified to have HCV viremia and were linked to care, please state how long were needed in the 4 patients to receive DAAs?

This information has now been added (results section 3.3, page 7):
“Three persons started DAA treatment in September 2020 (7 months after testing), two of whom completed treatment and one was lost to follow-up (he was incarcerated). One person initially refused treatment, but started in February 2021 (12 months after testing).”

2. Did all viremic patients receive baseline routine testing (hematological, biochemical etc…) before initiating DAAs? Because some DAAs are contraindicated for patients with decompensated liver diseases or not suitable for active HCCs, pre-treatment full assessments are needed for all treated patients. Can the authors provide the disease status for all the viremic patients and the information about the use of DAAs (which kind of DAAs, treatment duration)?

Given the hard-to-reach population, we decided to prescribe DAAs if the patient reported no contra-indications at intake. We only used one DAA combination, glecaprevir/pibrentasvir. A follow-up appointment was scheduled at the hospital 6 weeks later to test for liver function and to conduct an ultrasound and transient elastography measurement.

We have provided more information regarding contraindications and treatment (See section 2.3):
“Given the hard-to-reach population, we decided to prescribe DAAs if the patient reported no contra-indications at intake. DAA treatment included an 8-week course of glecaprevir/pibrentasvir.”

Regarding disease status of the patients; see section 3.3:

“During routine testing, this same participant showed suspicion of liver fibrosis based on the TEM and therefore follow-up testing of liver function was also scheduled in September. The other patient showed no liver damage during routine testing.”

3. Why did the patients receive testing after 6 weeks of treatment because it was not usual for this assessment? Please clarify it.

We acknowledge that this is not usually done. We thought that any medical issues of providing treatment sooner would outweigh the potential loss of follow-up, given that this is a hard-to-reach population.

4.Why did the patients receive HCV RNA testing after 8 weeks of treatment in the Methods? It was not correct to assess SVR during treatment. The testing should be done at 12 weeks off therapy. The illustration in Section 3.3 should also be clarified.

Our apologies for the unclear information. Information on the time points for HCV testing and treatment outcomes has been added to Section 3.3:

 “Of the two persons who completed treatment, SVR12 was assessed in one individual. For the other individual, EOT was delayed to 6 weeks after EOT due to limited testing during the corona crisis. We were unable to contact the participant for an appointment 12 weeks after EOT; however, an appointment is scheduled in September 2021 to assess SVR12. For both of these individuals, HCV RNA was negative at their last assessment.”

Reviewer 2 Report

In the manuscript entitled "A feasibility study to increase chronic hepatitis C virus RNA testing and linkage to care among clients attending homeless services in Amsterdam, the Netherlands", Ellen General and colloquies report the result of HCV sierology tests performed in homeless services in Amsterdam. They tested around 150 people and found 7 positive individuals, the majority of which accepted antiviral treatment. 

HCV is a major health issue worldwide, with around 70 million people chronically infected, at high risk of developing serious liver disease including cirrhosis, fibrosis steatosis and Hepatocarcinoma. In most cases  symptoms require decades to become evident and therefore most infected individuals are currently unaware of their status, thus being at high risk of transmitting infection to others. Among infected individuals intravenous drug users (IVU) are considered the most likely to transmit the infection and in many countries of the world have been given highest priority for treatment with recently approved direct actin antivirals (DAA). 

Indeed, approval of such highly effective drugs (>95 % SVR for all genotypes) has recently revolutionized HCV infection management, offering the possibility to incredibly reduce HCV prevalence and mortality worldwide. 

Keeping this in mind, the present work is interesting and useful. 

However, the methodology is not entirely clear, and results not completely clear, therefore I have a few concerns that need to be addressed before publication.

Mayor points

1) What test was used for screening patients for HCV RNA? Was it a validated test? What is the sensitivity? Such informations must be disclosed and discussed. A sensitivity below 95% would raise serious concerns.

2) What kind of antiviral regimen was proposed to (and given to some of the) positive individuals?

Minor comments

*)line 26 It is surprising to learn that one patient could not be treated because of drug interactions. With the large variety of DAA available it would be important to have more details in this respect

*) table I.

please define IQR

Please define "Dental treatment"

please explain the meaning of data shown under "Medical History of HCV). It looks like 7 individuals were previously known to be HCV+. This is in contrast with the claims of the authors that they identified four previously unknown HCV infected individuals (line 270)

*) Figure 1. It would be nice to have percentage values side by side to absolute values.

*) Line 218. "Three HCV positive..." this claim is in apparent contrast to data reported in Table 1 from which it appears that 7 clients tested positive to HCV test in the past.

*) Line 220. "declared that he was not treated in the past because of injecting drug use". This is an interesting point, since as far as this Reviewers knows, IVU are considered as "high priority" for DAA treatment, in that they represent an high risk of potential transmission. More information on the details of such treatment denial would be welcome. When did this happen? Where? 

*) Table 2. It would be important to also include data re: risk factors for positive clients individually. This could be represented with a matrix, with 7 columns (one per client) and 7 rows (one per risk factor) so that the reader could understand which risk factors were present in each patient. 

*) Line 250. What is the % of HCV positive individuals in people admitted to homeless services in the Netherlands, and in Europe?

*) Line 266. "90 euros". What is the price of the specific HCV test used in the study? Please include such information.

*) Line 324-326. An estimation of the costs of such strategy might be useful here, as compared to the alternative strategy (serology tests followed by RNA test of seropositive individuals)

*) Line 270 "4 of 7 were unaware of their status". This is in contrast with data shown in Table 1. Also, it would be nice to know how these numbers change if we consider IVU and non IVU. Similarly it would be nice to know if the "newly discovered" HCV+ individuals are IVU or not IVU.

*) Since the % of HCV positivity  is much higher in IVU+ vs IVU- it would be nice to show, beside aggregate data, also data obtained by keeping such categories separated.

Author Response

In the manuscript entitled "A feasibility study to increase chronic hepatitis C virus RNA testing and linkage to care among clients attending homeless services in Amsterdam, the Netherlands", Ellen General and colloquies report the result of HCV sierology tests performed in homeless services in Amsterdam. They tested around 150 people and found 7 positive individuals, the majority of which accepted antiviral treatment. 

HCV is a major health issue worldwide, with around 70 million people chronically infected, at high risk of developing serious liver disease including cirrhosis, fibrosis steatosis and Hepatocarcinoma. In most cases  symptoms require decades to become evident and therefore most infected individuals are currently unaware of their status, thus being at high risk of transmitting infection to others. Among infected individuals intravenous drug users (IVU) are considered the most likely to transmit the infection and in many countries of the world have been given highest priority for treatment with recently approved direct actin antivirals (DAA). 

Indeed, approval of such highly effective drugs (>95 % SVR for all genotypes) has recently revolutionized HCV infection management, offering the possibility to incredibly reduce HCV prevalence and mortality worldwide. 

Keeping this in mind, the present work is interesting and useful. 

However, the methodology is not entirely clear, and results not completely clear, therefore I have a few concerns that need to be addressed before publication.

Mayor points

1) What test was used for screening patients for HCV RNA? Was it a validated test? What is the sensitivity? Such information must be disclosed and discussed. A sensitivity below 95% would raise serious concerns.

This information has been added to the methods, page 3:
“We performed onsite HCV RNA testing using a validated blood fingerstick assay (Xpert HCV Viral Load assay, Cepheid, Sunnyvale, California, USA) [14, 15]. In a previous study, the sensitivity and specificity of this assay to detect HCV RNA in samples collected by finger-stick were 95.5% (95%CI 84.5-99.4) and 98.1% (95%CI 93.4-99.8), respectively [15].” 

2) What kind of antiviral regimen was proposed to (and given to some of the) positive individuals?

We have now added the following information:
Section 2.3: “DAA treatment included an 8-week course of glecaprevir/pibrentasvir.”

Minor comments

*)line 26 It is surprising to learn that one patient could not be treated because of drug interactions. With the large variety of DAA available it would be important to have more details in this respect

One HCV RNA positive individual was being treated for epilepsy. He was requested to contact his specialist to change this treatment, but refused.

We have now added the following information (Section 3.3):
“For one person, DAA treatment could not be started due to a potential drug-drug interaction with his anti-epileptic treatment. The participant did not wish to contact his medical specialist to change his treatment regimen.”

 *) table I.

please define IQR

IQR has been defined in the footnotes of Table 1 and at first mention in section 3.1.

Please define "Dental treatment"

We have changed the term “dental treatment” to “dental procedures”, as it includes any procedure performed by the dentist (regular checkups and more intense procedures, such as fillings, root canals, and crown/bridge placements).

please explain the meaning of data shown under "Medical History of HCV). It looks like 7 individuals were previously known to be HCV+. This is in contrast with the claims of the authors that they identified four previously unknown HCV infected individuals (line 270)

Thanks for noticing this discrepancy. We have corrected and clarified this section.

The reviewer may have confused prior and current infection. Regarding prior infection, seven individuals of the total sample (n=32) reported that they had tested HCV positive in the past, but only two of these seven individuals tested HCV RNA positive in our study. In our study, five persons of the 7 positive individuals in our study were unaware of their HCV status (the total of four persons mentioned in the original manuscript was incorrect). The remaining two HCV RNA positive individuals are those mentioned above.

We have added this information to the manuscript (page 5, section 3.1): “Seven persons (of n=32) reported that they had tested positive for HCV in the past (1985-2019), two of whom tested HCV RNA positive in our study.”  

We have added a row to Table 1 with the total amount of persons who had been previously tested for HCV (38/110). We also added a row indicating the number of people unaware of their HCV status for clarity.

See also footnote in Table 1: “Medical history was based on self-report, meaning that prior HCV testing could be based on both HCV antibody or RNA testing.”

*) Figure 1. It would be nice to have percentage values side by side to absolute values.

We have added percentages to Figure 1.

*) Line 218. "Three HCV positive..." this claim is in apparent contrast to data reported in Table 1 from which it appears that 7 clients tested positive to HCV test in the past.

Please refer to the previous response to Medical History.

*) Line 220. "declared that he was not treated in the past because of injecting drug use". This is an interesting point, since as far as this Reviewers knows, IVU are considered as "high priority" for DAA treatment, in that they represent an high risk of potential transmission. More information on the details of such treatment denial would be welcome. When did this happen? Where? 

We agree that current recommendations stress the need to treat persons who inject drugs. This patients was likely advised against treatment with interferon in 2010 due to active injecting drug use. This was standard practice at the time as patients needed to be motivated enough to complete 48 weeks of interferon.

We have added more details about the patient (page 7, section 3.2):  “The other person declared that he was not treated in 2010 because of injecting drug use. The medical specialist of the hospital likely advised against treatment with interferon at that time.”

*) Table 2. It would be important to also include data re: risk factors for positive clients individually. This could be represented with a matrix, with 7 columns (one per client) and 7 rows (one per risk factor) so that the reader could understand which risk factors were present in each patient. 

We added a matrix showing the presence HCV related risk factors and sexual behavioral factors for each HCV RNA positive individual: see supplementary table 5. We were not able to include the sociodemographic variables in this matrix to prevent identifiability of research participants.

*) Line 250. What is the % of HCV positive individuals in people admitted to homeless services in the Netherlands, and in Europe?

To our knowledge, there is only one previous study on HCV prevalence among homeless individuals in the Netherlands (ref. 12), and this study was already mentioned in the discussion:

“An earlier pilot study among clients of homeless services in Amsterdam showed a comparable proportion with positive HCV RNA tests of 4.4% (10/225) [12].”

We have also added HCV prevalence estimates of homeless individuals in Europe to the discussion:

“Previous estimations of HCV prevalence in homeless PWID across Europe are commonly based on HVC antibodies and range from 35% to 70% (21). One cohort study conducted in Europe among homeless individuals, prisoners and PWID showed a HCV RNA prevalence ranging from 9% to 43% (22).”

*) Line 266. "90 euros". What is the price of the specific HCV test used in the study? Please include such information.

We have added this information to page 3, section 2.3: “Assays (at €90 per test) and the testing device were provided free-of-charge by the supplier for the duration of this project”.

*) Line 324-326. An estimation of the costs of such strategy might be useful here, as compared to the alternative strategy (serology tests followed by RNA test of seropositive individuals)

We agree that this information would be useful; however, we do not know the total number of HCV antibody positive individuals in our study, hence any direct calculation of cost savings would not be possible.

*) Line 270 "4 of 7 were unaware of their status". This is in contrast with data shown in Table 1. Also, it would be nice to know how these numbers change if we consider IVU and non IVU. Similarly it would be nice to know if the "newly discovered" HCV+ individuals are IVU or not IVU.

This was an error. In fact, five out of seven HCV RNA positive individuals were newly discovered. Please refer to the comment above of reviewer 2.

We have also added more information on injecting drug use (page 7-8): “Of the 5 persons unaware of their HCV status, two reported recent and prior injecting drug use, two reported no recent or prior injecting drug use, and one did not answer this question during the interview.”

*) Since the % of HCV positivity  is much higher in IVU+ vs IVU- it would be nice to show, beside aggregate data, also data obtained by keeping such categories separated. 

This information can be directly calculated from Table 2. To save on word space, we decided not to report these specific percentages.

Round 2

Reviewer 2 Report

In the revised version the authors satisfied all concerns. However there a still a few issues before the manuscript can be publlished.

  • throughout the manuscript and in figures. "fingerprik" is probably misspelled.
  • line 66 "dropout... oral swabs". Please  provide details re: oral swabs performed.
  • were positive patients confirmed somehow, beside the use of the finger stick?
  • line 371 "This research was funded by Gilead Sciences and AbbVie B.V. Cepheid....", Is there a comma missing here?
  • line 371 "This research was funded by Gilead Sciences and AbbVie B.V. Cepheid....", I am not sure this does not represent a conflict of interest.
    •  

Author Response

Throughout the manuscript and in figures. "fingerprik" is probably misspelled.
This misspelling has now been corrected using ‘fingerstick’ throughout the manuscript.

line 66 "dropout... oral swabs".
Please provide details re: oral swabs performed. were positive patients confirmed somehow, beside the use of the finger stick?
We added a description of the test to line 66: “initial HCV rapid antibody test (OraQuick)”

line 371 "This research was funded by Gilead Sciences and AbbVie B.V. Cepheid....", Is there a comma missing here?
See funding, line 371: We changed this sentence: “This research was funded by AbbVie B.V. and Gilead Sciences. Cepheid provided the..”

line 371 "This research was funded by Gilead Sciences and AbbVie B.V. Cepheid....", I am not sure this does not represent a conflict of interest.
The reviewer is correct, we provided more details (see conflicts of interest):
“This project was funded by pharmaceutical companies, one of which provided the HCV tests and testing device. We declare that these companies were not involved in the study preparation, data collection, data analyses, data interpretation and the writing of this manuscript.”